# Identification of microRNAs That Provide a Low Light Stress Tolerance-Mediated Signaling Pathway during Vegetative Growth in Rice

**DOI:** 10.3390/plants11192558

**Published:** 2022-09-28

**Authors:** Sudhanshu Sekhar, Swagatika Das, Darshan Panda, Soumya Mohanty, Baneeta Mishra, Awadhesh Kumar, Devanna Basavantraya Navadagi, Rameswar Prasad Sah, Sharat Kumar Pradhan, Sanghamitra Samantaray, Mirza Jaynul Baig, Lambodar Behera, Trilochan Mohapatra

**Affiliations:** 1Crop Improvement Division, ICAR-National Rice Research Institute (NRRI), Cuttack 753006, India; 2Former Secretary DARE, DG, ICAR, Government. of India, New Delhi 11001, India

**Keywords:** index: low light, miRNA, rice, Swarnaprabha, IR8, vegetative stage, chlorophyll

## Abstract

Low light intensity affects several physiological parameters during the different growth stages in rice. Plants have various regulatory mechanisms to cope with stresses. One of them is the differential and temporal expression of genes, which is governed by post-transcriptional gene expression regulation through endogenous miRNAs. To decipher low light stress-responsive miRNAs in rice, miRNA expression profiling was carried out using next-generation sequencing of low-light-tolerant (Swarnaprabha) and -sensitive (IR8) rice genotypes through Illumina sequencing. Swarnaprabha and IR8 were subjected to 25% low light treatment for one day, three days, and five days at the active tillering stage. More than 43 million raw reads and 9 million clean reads were identified in Swarnaprabha, while more than 41 million raw reads and 8.5 million clean reads were identified in IR8 after NGS. Importantly, 513 new miRNAs in rice were identified, whose targets were mostly regulated by the genes involved in photosynthesis and metabolic pathways. Additionally, 114 known miRNAs were also identified. Five novel (*osa-novmiR1*, *osa-novmiR2*, *osa-novmiR3*, *osa-novmiR4*, and *osa-novmiR5*) and three known (*osa-miR166c-3p*, *osa-miR2102-3p*, and *osa-miR530-3p*) miRNAs were selected for their expression validation through miRNA-specific *q*RT-PCR. The expression analyses of most of the predicted targets of corresponding miRNAs show negative regulation. Hence, miRNAs modulated the expression of genes providing tolerance/susceptibility to low light stress. This information might be useful in the improvement of crop productivity under low light stress.

## 1. Introduction

Light intensity plays a central role in plants obtaining energy for their survival [1]. Low light stress is a problem during the wet (rainy) season around the world, especially in Asian countries, where most of the population depends on rice production for their food. Due to low light intensity, several physiological parameters are changed both at the vegetative and reproductive stages. It exerts negative impacts on photosynthetic activity and ultimately hampers grain yield and quality [2] in crop plants. Fluctuating light intensities disturb the photosynthetic machinery in thylakoid membranes [3]. Vegetative growth and development in plants are determined by cell division, elongation, and branching [4]. Light is an important environmental parameter that can regulate it [5]. Plants evolved sophisticated photoreceptors that control many biochemical and physiological parameters to increase their photosynthetic and metabolic performance by adapting to different light environments [6,7]. The photosynthetic rate is considerably reduced under low light compared to normal light. Light intensity plays the main role in germination, photosynthesis, leaf proliferation and expansion, and cell division in plants [8,9]. Low light intensity may affect the carbon balance of crop plants as the demand for carbohydrates increases and their production decreases due to low photosynthetic yield under shade stress [10]. Low light intensity during the grain-filling stage causes decreases in the supply of assimilates to the grains, resulting in a decrease in starch synthase activity, which leads to low grain filling and chalkiness of rice [11]. The reduction in carbohydrate availability under low light was responsible for the delayed flowering [12] in Swarnaprabha rice [13], and some soybean lines [14]. Low light stress reduced inflorescence, delayed flower bud differentiation and shortened flowering time [13,14]. Due to low light intensity, crop plants have thinner and smaller leaves than leaves in normal light [15]. However, under low light, plant height and lodging rate were increased, which decreased the transportation of nutrients, water, and photosynthetic assimilates, leading to the loss of agriculture production [16]. Low light usually induces a hyponastic response in plants [17,18,19] and is controlled by cryptochrome and phytochrome [20,21,22]. Therefore, due to low light, plants absorb more light and increase carbon gain under this condition [23,24]. In addition, light is the only source of energy for starch biosynthesis [25]. The starch biosynthesis and degradation are adjusted by the availability of sunlight, and it enhanced starch synthesis when the availability of light decreased the rate of degradation [26]. Under the reduced light intensity, starch content was reduced in bean leaves and sugar beet (*Beta vulgaris*) [27,28]. Tolerant rice genotypes Purnendu, Swarnaprabha, and Sashi show efficient shade-tolerant mechanisms with sustainable yield under shade [29,30]. These genotypes can maintain higher yields as compared to susceptible rice genotypes under shade by maintaining higher net photosynthesis rates, better translocation efficiencies from source to sink organs, nutrient remobilization for growing organs, the recycling of chloroplast and root plastids, higher grain numbers, higher panicle numbers, and higher rates of panicle emergence and antioxidant activities to restrict membrane damage [14,31,32,33]. The tolerant genotypes Pantdhan19, Purnendu, and Sashi show less yield reduction under shade by maintaining or even increasing grain numbers and grain weight [30].

During the course of evolution, plants have developed numerous regulatory mechanisms to cope with stress, one of them being the differential and temporal expression of genes, which is governed by post-transcriptional gene expression regulation through miRNAs. miRNAs are a group of endogenous small RNAs (20–24 nucleotide) that have important gene-regulatory roles through base pairing with target mRNAs for cleavage or translational inhibition [34]. Numerous studies have been conducted involving miRNAs in the regulation of plant growth development and yield-related traits in rice. MiRNAs are broadly distributed throughout the plant kingdom and evolutionarily conserved in monocots and dicots, such as rice, *Arabidopsis,* and cotton [35]. miRNAs have a role in the regulation of growth and development processes that include growth from vegetative to the reproductive stage [36,37], root development [38], and leaf morphogenesis [39]. Due to abiotic stresses, the expression levels of miRNAs change, resulting in a modulation of the expression patterns of miRNA target genes that are associated with stress adaptations [40] as well as providing the plants with genome stability. The regulatory actions of miRNAs have been demonstrated in abiotic stresses such as low and high temperature, drought, salinity, heavy metal, phosphate deficiency, etc., in different plant species including rice, so as to ensure sustainable agricultural production [41,42,43,44,45,46,47]. The developments in CRISPR-Cas9 technologies have given the idea that miRNAs can serve as a crucial tool for the improvement of several important agricultural traits [48,49]. A better understanding of the regulation of miRNAs and targets during stress responses can contribute to rice breeding for improving yield, quality, and tolerance to abiotic stresses [40]. miRNAs and their targets are involved in the phytochrome-B mediated light-signaling pathway, which regulates the expression of transcription factors. A number of miRNAs, *miR164*, *miR166*, *miR167*, *miR168*, *miR169*, *miR530*, and *miR2879*, have been identified that are light-inducible [50]. *miR156,* the most evolutionarily conserved miRNA, was found to control the expression of SQUAMOSA promoter-binding protein-like (*IPA1/OsOSPL14*) genes in *Arabidopsis* [51] and rice [52,53]. The *miR156-SPLs* module acts as a negative regulator of the shade-avoidance syndrome, an adaptive strategy of plants to avoid shade from the canopy or compete for light with their neighbors [54]. In *Arabidopsis*, the low light intensity can activate phytochrome interacting factors that directly bind to and transcriptionally repress the expression of *miR156s,* leading to the upregulation of *IPA1*/*OsOSPL14*. *IPA1*/*OsOSPL14* mediates diverse morphological changes that are associated with enhanced shade-avoidance syndrome responses [55]. In rice, *miR2118* targets a long noncoding RNA, Photoperiod-sensitive Genic Male Sterlity1 Transcript (*PMS1T*)-producing phased small-interfering RNAs, and regulates photoperiod-sensitive male sterility [49]. During the seedling de-etiolation as well as seed germination in *Arabidopsis, miR163* is highly induced by light [56]. Under three different light conditions—control (200 μmol m^−2^ s^−1^, 4 h), high intensity (1200 μmol m^−2^ s^−1^, 4 h), and dark (dark, 24 h)—miRNA expressions were evaluated in *Dendrocalamus latiflorus* [57] and found that expression of *miRC18*, *miRC27-5p*, and *miRC27-3p* significantly increased, whereas *miRC19* and *miRC28* expressions were reduced under the high light intensity. Similarly, under dark conditions, *miRC1*, *miRC22*, *miRC25*, *miRC27- 5p*, and *miRC27-3p* were upregulated, whereas *miRC5*, *miRC17*, and *miRC29* were downregulated significantly [57]. This evidence suggests that miRNAs play a vital role in the regulation of genes involved in light stress-mediated cellular responses [57]. Hence, miRNAs are the key components of gene regulation under abiotic stresses. During the reproductive stage, it was reported that under prolonged low light stress, *miR5493, miR5144, miR5493, miR6245*, *miR5487*, *miR168b*, *miR172b*, and *miR168b* were expressed, which regulate the expression of their targets *OsSLAC*, *OsLOG1*, *OsBRITTLE1-1*, *OsCsIF9*, *OsGns9*, *OsbHLH153*, *OsCP1*, and *OsDET1*, respectively in tolerant rice genotype, Swarnaprabha [58]. Panigrahy et al. [58] observed the upregulation of most of the ethylene and cytokinin pathway genes, including *ETHYLENE-RESPONSIVE BINDING PROTEIN 2 (EREBP2)*, *MOTHER OF FLOWERING TIME 1 (MFT1),* and *SHORT PANICLE 1 (SP1)* in shade-grown panicles of Swarnaprabha. Further, they demonstrated the role of these miRNAs in prolonged-shade tolerance during the reproductive stage, which significantly contributed to the shade-tolerance phenotypes in the shade-tolerant rice variety Swarnaprabha. These phenotypes include better pollen development (*miR5487-OsGns9* and *miR168b-OsCP1*), grain formation (*miR5493-OsBRITTLE1-1*), enhanced panicle size (*miR5493-OsSLAC* and *miR5144-OsLOG1*), hyponasty (*miR172b-OsbHLH153*), etc., under shade. To date, no low-light stress-responsive miRNAs have been identified to understand the low-light-mediated signaling pathways during vegetative growth in rice.

To obtain a proper understanding, concerted efforts on studies on miRNAs and their target mRNAs are needed. Hence, to identify known and novel miRNAs and their targets involved in the low light stress-mediated signaling in rice, we performed high-throughput small RNA sequencing and their differential expression analysis of normal and low-light-treated samples of low-light-tolerant and -sensitive rice genotypes at the active tillering stage. A total of eight small RNA libraries were generated from leaf tissues of low-light-tolerant and -sensitive rice genotypes after 25% low light treatment. Rice genotypes Swarnaprabha and IR8 were taken as low-light-tolerant and -sensitive rice genotypes, respectively, based on earlier reports [59,60,61,62,63]. The Illumina HiSeq 2500 platform was used for sequencing the prepared library. Differentially expressed miRNAs between samples were identified and their targets were predicted. Importantly, both novel and known miRNAs were identified through sequencing, whose targets were mostly involved in photosynthesis and metabolic pathways. Five novel and three known miRNAs were validated using miRNA-specific *q*RT-PCR. Further, their expressions were validated in two more tolerant (Purnendu, VLDhan-209) and one sensitive (GR4) rice genotypes along with Swarnaprabha and IR8 in the next season. The low-light-tolerant and -sensitive rice genotypes were selected based on the previous reports and our previous experiments [30,64,65].

## 2. Results

### 2.1. Physiological Parameters of Tolerant and Sensitive Rice Genotypes

PAR values under LL and NL conditions above the canopy of rice genotypes Swarnaprabha and IR8 were recorded during the *Kharif* season of 2018 (Appendix A). Similarly, PAR values under LL and NL conditions above the canopy of rice genotypes Swarnaprabha, VLDhan-209, Purnendu, GR4, and IR8 were recorded during the *Kharif* season of 2019 at three times of the day (9.00 am, 12.00 noon, and 4.00 pm Indian Standard Time (IST) after LL treatment to confirm the LL and NL intensity (Appendix A). Compared to NL, there was a ~25% significant (*p* < 0.05) reduction of sunlight under the LL treatment. Maximum PAR was recorded at noon (mean NL: 1200 μmol m^2^ s^−1^; LL: 900 μmol m^2^ s^−1^) followed by a gradual depletion until the late afternoon hours (4.00 pm). After 5 days of low light treatment, the chlorophyll a and chlorophyll b content of tolerant and sensitive rice genotypes were recorded. The chlorophyll a content was significantly increased in all the genotypes under low light at *p* < 0.05 (*t*-test for mean at α = 0.05), but chlorophyll b content was significantly increased in tolerant genotypes Swarnaprabha, VLDhan-209, and Purnendu at *p* < 0.01, and therefore the chlorophyll a/b ratio (Figure 1A,C) was significantly decreased (*p* < 0.01, *t*-test for mean at α = 0.01). However, the chlorophyll b content was not significantly increased in sensitive genotypes GR4 and IR8. Interestingly, the rate of reduction in Chl a/b in Swarnaprabha, VL Dhan-209, and Purnendu was higher (13–28%) than that in GR4 and IR8 (3–6%). Further, physiological parameters such as net assimilation, transpiration, and stomatal conductance were estimated in all the genotypes under normal and low-light conditions (Figure 1B,D) in both the *Kharif* seasons of 2018 and 2019. Results indicated that low-light-tolerant genotypes maintained significantly (*p* < 0.01) higher values in net assimilation, transpiration, and stomatal conductance compared to low-light-sensitive rice genotypes, though there were reductions under low light compared to the normal light condition in all the genotypes. The mean percentages of reduction of net assimilation, transpiration, and stomatal conductance under LL in Purnendu, VL Dhan 209, Swarnaprabha, GR4, and IR8 were 11%, 19%, 23%, 55%, 58%; 11%, 18%, 24%, 71%, 67% and 19%, 20%, 22%, 59%, 5%, respectively.

### 2.2. Small RNA Sequencing of Low-Light-Tolerant and -Sensitive Rice Genotypes

To identify miRNAs involved in low-light-mediated signaling, a total of eight small RNA libraries were prepared: four from the tolerant genotype Swarnaprabha and another four from the sensitive rice genotype IR8 for one day, three days, and five days, low-light-treated along with control samples and further subjected to small RNA sequencing using the Illumina HiSeq 2500 NGS platform. A total of 41.38, 48.02, 42.37, and 43.51 million raw reads were generated for control, T1, T3, and T5 samples of Swarnaprabha, respectively, whereas 47.13, 50.06, 33.18, and 34.54 million raw reads were generated for control, T1, T3, and T5 samples of IR8, respectively (Table 1). Further, raw reads were filtered through cutadapt software using the parameter having a read length of 18 to 24 nt and low-quality (Q20%) bases. A total of 10.43, 9.25, 9.72, and 7.85 million clean reads for Swarnaprabha and 11.17, 10.42, 8.64, and 4.93 million clean reads for IR8 were identified for the samples control, T1, T3, and T5, respectively (Table 1). The abundance of miRNAs was higher in the control samples, whereas after low light treatment, clean reads decreased in both tolerant and sensitive rice genotypes (Table 1). The distribution of read count of miRNAs for Swarnaprabha and IR8 shows that the degree of redundancy decreased in treated samples from 18 to 24 nt as compared to the control sample in IR8. However, in Swarnaprabha, the read count of the miRNAs’ sequence was decreased from 18 to 21 nt after treatment, but for 23 and 24 nt, the read length was increased in T3. However, these were comparatively lower in the T5 samples but higher as compared in the T1 samples (Figure 2). Further, 22 to 24 nt-length miRNAs were most abundantly found in the overall miRNA sequencing data (Figure 2).

### 2.3. Identification of miRNAs

After stringent scrutiny and filtration of read counts as mentioned in the Materials and Methods section, a total of 95, 116, 120, and 104 novel and 53, 61, 72, and 59 known miRNAs were identified, respectively, in the control, T1, T3, and T5 samples of sensitive rice genotype IR8, whereas a total of 127, 130, 107, and 58 novel and 68, 61, 72, and 52 known miRNAs were identified, respectively in the control, T1, T3, and T5 samples of tolerant genotype Swarnaprabha (Appendix A).

### 2.4. Analysis of Differentially Expressed miRNAs

A considerable number of differentially expressed miRNAs were identified under low light stress among the samples. In total, 114 known miRNAs and 513 novel differentially expressed miRNAs were identified in tolerant and sensitive rice genotypes from eight prepared small RNA libraries based on the miARma-Seq tool Novel miRNA pipeline using miRDeep2 in miARma-Seq (Appendix A). A total of 88, 101, and 85 miRNAs were differentially expressed in the T1, T3, and T5 samples of sensitive rice genotype IR8, among which 53, 62, and 53 were upregulated, while 35, 39, and 32 were downregulated, respectively, as compared to the control sample (Table 2 and Appendix A). However, in the tolerant genotype Swarnaprabha, a total of 67, 65, and 83 miRNAs were differentially expressed in the T1, T3, and T5 samples, among which 33, 28, and 26 were upregulated, while 34, 37, and 57 were downregulated, respectively, as compared to the control sample (Table 2 and Appendix A). Hence, the data show that more miRNAs were differentially expressed in the sensitive genotype as compared to the tolerant genotype due to low light stress. Further, upregulated miRNAs were more abundant in the sensitive rice genotype as compared to downregulated miRNAs and vice versa, while the opposite was observed in the tolerant genotype (Table 2). In comparison between the samples of control, T1, T3, and T5 of the tolerant and sensitive rice genotypes, a total of 81, 94, 85, and 100 miRNAs were found differentially expressed, respectively, in which 48, 44, 33, and 30 were upregulated, while 33, 50, 52, and 70 were downregulated, respectively. The comparison between Swarnaprabha (S) and IR8 (I) indicated that the number of downregulated miRNAs increased after low light treatment from 1 day to 5 days in the tolerant genotype Swarnaprabha, whereas the number of upregulated miRNAs decreased. Reverse trends were observed in the sensitive rice genotype IR8. The clustering of the 250 most differentially expressed miRNAs of all samples was denoted by the heat map diagram using the log_2_ (FPKM + 1) value shown in Figure 3. The green color indicated downregulated miRNAs, whereas the red color showed upregulated miRNAs. A Venn diagram of differentially expressed miRNAs within and between the samples of tolerant and sensitive rice genotypes was shown in Figure 4. The sum of numbers in each circle is the total number of differentially expressed miRNAs within a group, and the overlap represents the miRNAs expressed commonly in between comparisons. Only 15 miRNAs differentially expressed in response to low light treatment were found to be common in Swarnaprabha (Figure 4a). In contrast, the number of common differentially expressed miRNAs in response to low light stress in IR8 was 25 (Figure 4b) as compared to the control. Only 14 differentially expressed miRNAs were common among the different comparisons between the Swarnaprabha and IR8 genotypes (Figure 4c). At different stages of low-light-treated samples, a total of 22, 16, and 24 miRNAs were specifically expressed in the T1, T3, and T5 samples of the tolerant genotype Swarnaprabha, whereas a total of 18, 31, and 20 miRNAs were only expressed in the T1, T3, and T5 samples of the sensitive genotype IR8, respectively (Figure 4a,b).

### 2.5. Expression Validation of Differentially Expressed miRNA NGS Data in Tolerant and Sensitive Rice Genotypes through qRT-PCR

Based on the role of miRNA targets in photosynthesis and metabolic pathways, differentially expressed miRNAs were identified and selected for their expression validation in both tolerant and sensitive rice genotypes. A total of three known and five novel differentially expressed miRNAs were selected for their expression validation through *q*RT- PCR. The expression patterns of three known miRNAs, *osa-miR166c-3p, osa-miR2102-3p,* and *osa-miR530-3p,* were validated in low-light-treated samples as compared to the control samples of Swarnaprabha and IR8 (Figure 5). miRNA-specific quantitative real-time PCR was used for their expression validation. For *osa-miR166c-3p,* both NGS and *q*RT-PCR data showed that its expression was upregulated in the tolerant genotype Swarnaprabha while downregulated in the sensitive rice genotype IR8 in 1 day, 3 days, and 5 days of low-light-treated samples (Figure 5A). *osa-miR2102-3p* and *osa-miR530-3p* were downregulated in Swarnaprabha while upregulated in IR8 as compared to the control (Figure 5B,C) except in the T1 sample of Swarnaprabha, where *osa-miR530-3p* was upregulated. The expression of five novel differentially expressed miRNAs was also validated through quantitative PCR in the tolerant and sensitive rice genotypes in a similar way to known miRNAs. The expression of novel miRNA *osa-novmiR1* was upregulated in the tolerant rice genotype Swarnaprabha except in T3 samples, whereas it was downregulated in IR8 (Figure 6A). The expression of novel miRNA *osa-novmiR2* was downregulated in both the tolerant and sensitive rice genotypes (Figure 6B) except in the T1 sample of Swarnaprabha (Figure 6B). The expression of novel miRNA, *osa-novmiR3* was upregulated in Swarnaprabha while downregulated in IR8 (Figure 6C). The expressions of novel miRNA, *osa-novmiR4*, and *osa-novmiR5* were downregulated in the low-light-tolerant genotype Swarnaprabha while upregulated in the sensitive rice genotype IR8 as compared to the control (Figure 6D,E) sample, except for the expression of *osa-novmiR4* in the T1 sample of IR8.

The expression validation of all above eight identified miRNA was also performed in two more tolerant Purnendu, VLDhan-209, and one sensitive GR4 rice genotype along with Swarnaprabha and IR8 in the next season (*Kharif* 2019) to further confirm the results. The results are shown in Appendix A. The expression pattern of known miRNA *osa-miR166c-3p* was also upregulated in the tolerant genotypes Purnendu, VLDhan-209, and Swarnaprabha, whereas it was downregulated in the sensitive rice genotypes GR4 and IR8, except for the 1-day sample of GR4 (Appendix A). *osa-miR2102-3p* was downregulated in all three of the tolerant genotypes while upregulated in the sensitive rice genotypes, except for the 1-day sample of IR8 (Appendix A). Further, the expression of known miRNA *osa-miR530-3p* was also downregulated in all the tolerant rice genotypes except for 1-day LL-treated samples, whereas it was upregulated in the sensitive rice genotypes (Appendix A). The expression pattern of novel miRNA in the sample of *Kharif* season 2019 was also more or less similar to the samples of *Kharif* season 2018 in Swarnaprabha and IR8. The expressions of novel miRNA *osa-novmiR1* and *osa-novmiR3* were upregulated (Appendix A) in tolerant rice genotypes, whereas they were downregulated in the sensitive rice genotypes, except for the 1-day LL-treated samples. The expressions of novel miRNA osa*-novmiR4* and *osa-novmiR5* were downregulated in the tolerant rice genotypes, whereas they were upregulated in the sensitive rice genotypes (Appendix A). The expression pattern of novel miRNA osa*-novmiR2* was downregulated in both the tolerant and sensitive rice genotypes, except in the 1-day LL-treated samples of tolerant rice genotypes, but the expression was higher in the tolerant rice genotypes as compared to the sensitive rice genotypes (Appendix A).

### 2.6. Expression Analysis of miRNAs’ Target Genes through qRT-PCR

To understand the biological role of the known/novel miRNAs differentially expressed in the tolerant and sensitive rice genotypes, their putative target genes were searched using the psRNATarget tool. The expression analysis of targets of three known and five novel miRNAs was used for their expression study. The most predicted target gene with an expectation value of 1 and UPE (unpaired energy) value of 22.59 for *osa-miR166c-3p* is rolled leaf1, which is also known as homeobox-leucine zipper protein HOX9-like protein. Its expression was downregulated as compared to control in the tolerant genotype Swarnaprabha, whereas it was the reverse in the sensitive genotype IR8. Hence, the expression of miRNA *osa-miR166c-3p* has a negative correlation with its target rolled leaf1’s expression in the tolerant and sensitive genotypes, as it suppresses the expression of rolled leaf1 through cleavage (Figure 7A). The predicted target of *osa-miR2102-3p* with an expectation value of 2.5 and an UPE value of 22.194 is chlorophyll a-b-binding protein. Its expression was found upregulated in the tolerant genotype Swarnaprabha, whereas it was downregulated in the sensitive rice genotype IR8 (Figure 7B). The expression of miRNA osa*-miR2102-3p* was downregulated in the tolerant rice genotype while upregulated in the sensitive genotype (Figure 5B). For the known miRNA *osa-miR530-3p,* the predicted target, ubiquinone biosynthesis protein COQ4, with an expectation value of 3.5 and UPE value of 21.288, was selected for expression analysis. Its expression was significantly downregulated in IR8 but nonsignificantly downregulated in the tolerant genotype Swarnaprabha (Figure 7C). The most predicted target of novel miRNA *osa-novmiR1* is ribonucleoside-diphosphate reductase, with an expectation value of 2.5 and UPE value of 21.371. Its expression was upregulated in both Swarnaprabha and IR8 after low light treatment (Figure 8A). The target gene for novel miRNA *osa-novmiR2* (expectation value of 2, UPE value of 21.35) is cytochrome P450 87A3. Its expression was upregulated in IR8, whereas it was downregulated in the T1 sample of Swarnaprabha while upregulated in the T3 and T5 samples of both IR8 and Swarnaprabha after low light treatment (Figure 8B). The predicted target gene of novel miRNA *osa-novmiR3* is granule-bound starch synthase 1b (LOC_Os07g22930.3), which is also known as granule-bound starch synthase II with an expectation value of 3.5 and UPE value of 17.32. Its expression was downregulated in both the tolerant (Swarnaprabha) and sensitive (IR8) rice genotypes as compared to the control sample (Figure 8C), but the expression was higher in IR8. The predicted target gene of novel miRNA, *osa-novmiR4*, is an NAC domain-containing protein with an expectation value of 2 and UPE value of 19.39. Its expression was upregulated in the tolerant genotype Swarnaprabha, whereas it was downregulated in the sensitive rice genotype IR8 as compared to its control (Figure 8D). The predicted target of novel miRNA *osa-novmiR5* is FAD-dependent oxidoreductase (expectation value of 3.5, UPE value) and cryptochrome2 apoprotein (expectation value of 3.5, UPE value of 23.23). The expressions of both FAD-dependent oxidoreductase and cryptochrome2 apoprotein were upregulated in the tolerant genotype Swarnaprabha but downregulated in the sensitive rice genotype IR8 as compared to the control sample (Figure 8E,F).

## 3. Discussion

Low light is a major abiotic stress that affects plant growth development and significantly reduces the yield potential of crops. Several physiological and biochemical parameters change upon low light stress. Plants suffer due to low light stress throughout the life cycle, but the tolerant high-yielding genotype shows less than 20–30% yield reduction under low light [30]. Little work has been performed to date to understand the low-light-mediated stress mechanisms in rice at the molecular level. In our previous work [63] a comparative transcriptome work of low-light-tolerant and -sensitive rice genotypes was performed to understand the low-light stress mechanism. In similar ways, we carried out several physiological analyses in more tolerant and sensitive rice genotypes in the present study. Here, we used two seasons’ data for the studies. In *Kharif* 2018, Swarnaprabha and IR8 were used for data collection and validation, whereas in *Kharif* season 2019, a total of three tolerant rice genotypes Swarnaprabha, VLDhan-209, and Purnendu, and two sensitive rice genotypes GR4 and IR8 were taken based on their previous report [30,64,65]. PAR values (Appendix A), chlorophyll a and b content, net assimilation, transpiration, and stomatal conductance were estimated for all these genotypes, like our previous work [63], to differentiate the tolerant and sensitive rice genotypes (Figure 1A–D). PAR values showed a significant difference between LL and NL treatments in both the *Kharif* seasons. The chlorophyll a content was significantly increased in all the genotypes under low light, but the chlorophyll b content was significantly increased in the tolerant genotypes Swarnaprabha, VLDhan-209, and Purnendu, and therefore the chlorophyll a/b ratio was significantly decreased. However, the chlorophyll b content was not significantly increased in the sensitive genotypes GR4 and IR8, indicating that low-light-tolerant genotypes increase their chlorophyll b content to a greater extent under low light in order to capture solar energy as per requirements and maintain the photosynthetic process even under low-light conditions. Other physiological parameters such as net assimilation, transpiration, and stomatal conductance were reduced significantly in low-light-sensitive rice genotypes as compared to tolerant genotypes under low light. Similar results were also observed in some of the low-light-tolerant and -sensitive rice genotypes [65].

miRNAs play a central role in the regulation of gene expression induced by abiotic stresses to cope with stresses. Next-generation sequencing using the Illumina platform is a rapid and high-throughput approach for identifying novel and known differentially expressed miRNAs and their targets in various plant species [66,67,68]. Small RNAs, particularly endogenous miRNAs in plants, have a very important role in regulation through controlling the expression of particular genes involved in the pathways. Panigrahy et al. [58] demonstrated the role of miRNAs in prolonged-shade tolerance in the shade-tolerant rice variety Swarnaprabha during the reproductive stage. They identified 16 new miRNAs with their 21 targets, which significantly contribute to the shade-tolerance phenotype in Swarnaprabha. These phenotypes include better pollen development, grain formation, enhanced panicle size, etc., under shade.

In the present study, 25% low-light-treated tolerant (Swarnaprabha) and sensitive (IR8) rice genotypes (at the active tillering stage) with three-time points of 24 h (T1), 48 h (T3), and 72 h (T5) with the control samples were collected. A total of eight small RNA libraries were constructed from these collected samples. Through Illumina sequencing (NGS), several differentially expressed miRNAs were identified. This is the first report to identify miRNAs in response to low light stress at the vegetative stage in rice. miRNAs having read lengths between 21 to 24 nt are involved in gene regulation through translation inhibition and mRNA cleavage [69]. In our study, clean read data of miRNAs mostly belonged to those 21 to 24 nt in length, particularly in the low-light-sensitive rice genotype IR8 control and T1 samples, but their value was diminished after 3 days and 5 days of low light treatment. In the tolerant genotype Swarnaprabha, its value was increased in the T3 samples and was comparatively more in number in T5 as compared to the sensitive rice genotype (Figure 2), indicating that after prolonged low light stress, the regulation of the gene through miRNA mediation was higher in the tolerant genotype as compared to the sensitive genotype to cope with stress. Differentially expressed miRNA data show that upregulated miRNAs were more abundant in IR8 as compared to downregulated ones and vice versa, while the opposite was seen in Swarnaprabha (Table 2). miRNA regulates the genes or transcripts by suppressing their expression through cleavage or translational inhibition. Hence, the expression of the gene (target) was suppressed more in the sensitive genotype as compared to the tolerant one with their respective control sample, and this enables the tolerant genotypes to maintain their photosynthetic activity and metabolic pathways even under low light. The Venn diagram shows complex scenarios of differentially expressed miRNAs among and in between tolerant and sensitive rice genotypes (Figure 4). Common miRNAs were more abundant in IR8 as compared to Swarnaprabha at different time point treatments (T1, T3, and T5) with respect to the control. In our study, several known and novel differentially expressed miRNAs were identified in the tolerant and sensitive rice genotypes (Table 2 and Appendix A) that may have a regulatory role in the low-light-mediated responsive pathways. 

We carried out expression analysis of three known miRNAs—*osa-miR166c-3p, osa-miR2102-3p,* and *osa-miR530-3p—*and five novel differentially expressed miRNAs—*osa-novmiR1, osa-novmiR2, osa-novmiR3, osa-novmiR4*, and *osa-novmiR5*. The expression of miRNA *osa-miR166c-3p* was upregulated in the tolerant while downregulated in the sensitive rice genotypes (Figure 5A and Appendix A). Its most predicted target gene is rolled leaf1. Nagasaki et al. [70] reported that homeobox-leucine zipper protein HOX9-like protein (rolled leaf1) expression was suppressed by *osa-miR166c.* In the tolerant genotype Swarnaprabha, its expression was downregulated upon low light treatment, whereas it was upregulated in the sensitive rice genotype IR8 (Figure 7A), indicating that miRNA *osa-miR166c-3p* has a negative correlation with the expression of rolled leaf1. Hence, it might be playing a role in regulating the low-light-mediated tolerance mechanisms in rice. *MiRNA166* expression during leaf growth and development, and further, their accumulation in the phloem, indicate that due to its expression, a movable signal may form that emanates from a signaling center below the incipient leaf [71] in maize. Chen et al. [72] also reported that an *Oryza sativa* dominant mutant, rolled, and erect leaf 1 (*rel1*) has a predominant rolled leaf, increased leaf angle, and reduced plant height phenotype that results in a reduction in grain yield. Hence, rolled leaf1 negatively regulates leaf morphology and plant growth development. *Osa-miR2102-3p* expression was found downregulated in the tolerant rice genotypes while upregulated in the sensitive rice genotypes (Figure 5B and Appendix A). Its predicted target is chlorophyll a-b binding protein (CAB). CAB expression was found upregulated in the tolerant genotypes after low light treatment whereas downregulated in the sensitive genotypes (Figure 7B) as compared to the control, indicating that it has also a negative correlation with its predicted target. Sekhar et al. [63] mentioned that the LHC (light-harvesting complex) has CAB (chlorophyll a-b binding protein) as part of the antenna protein complex, which functions as a light receptor that receives and transfers excitation energy to photosystem II and I [73]. The expression of CAB is regulated by multiple environmental factors. Light intensity is one of them [74]. The expression and abundance of CAB consistently change in response to light intensity [75,76]. Therefore, *osa-miR2102-3p* has a very important role in regulating the low-light-mediated response in rice. The predicted target gene of another known miRNA *osa-miR530-3p* is ubiquinone biosynthesis protein COQ4. The expression of *osa-miR530-3p* was found downregulated in the tolerant rice genotypes (Swarnaprabha, Purnendu, and VLDhan 209) except for the T1 sample while upregulated in T3 and T5 sample of the sensitive rice genotypes (IR8 and GR4) (Figure 5C). Its predicted target, ubiquinone biosynthesis protein COQ4, was downregulated in both Swarnaprabha and IR8 as compared to the control sample (Figure 7C). However, the downregulation in the tolerant genotype was not significant as compared to the control, except for the T1 sample (Figure 7C). Hence, it has a higher expression in the tolerant rice genotype Swarnaprabha as compared to the sensitive rice genotype IR8. Ubiquinone biosynthesis protein COQ4 synthesizes ubiquinone protein, which is an important prenylquinone and functions as an electron transporter in plants. In chloroplast thylakoids and mitochondrial inner membranes, it involves photophosphorylation and oxidative phosphorylation [77]. Higher production of this protein is required under stress conditions that can synthesize higher-energy molecules to cope with it.

The expression of novel miRNA *osa-novmiR1* was upregulated in the tolerant rice genotypes while downregulated in the sensitive rice genotypes (Appendix A). Its predicted target is ribonucleoside-diphosphate reductase (expectation value of 2.5), which is also known as ribonucleotide reductase (RNR). It regulates the rate of deoxyribonucleotide production in the pathway of DNA synthesis and repair [78]. RNR has two components, small and large subunits, and is highly expressed in shoot base and young leaves [78]. For chlorophyll synthesis and plant growth development, a small subunit of ribonucleotide reductase is required [79]. We found through physiological data that chlorophyll a and b contents increased after low light treatment at a significant level and the expression of RNR1 was also upregulated in both the tolerant and sensitive rice genotypes (Figure 8A) after low light treatment as compared to the control sample. Hence, RNR might be involved in low-light-mediated regulation to adapt to low-light stress. Novel miRNA *osa-novmiR2* shows higher expression in the tolerant rice genotypes as compared to the sensitive rice genotypes but is downregulated in both the tolerant and sensitive rice genotypes after low light treatment (Appendix A). Its most predicted target is cytochrome P450 87A3 (expectation value of 2), whose expression was found to be upregulated in IR8 while downregulated in the T1 sample of Swarnaprabha but upregulated in the T3 and T5 low-light-treated samples (Figure 8B). Cytochrome P450 87A3 transcript level was transiently upregulated in response to light in rice coleoptiles [80] and induced by auxin. Its expression remained stable or slightly increased after low light irradiance but disappeared after prolonged treatment in rice coleoptiles. CYP87A3 protein most likely has a negative regulator for the auxin responsiveness of growth [80]. Hence, irradiation might lead to auxin-independent gene regulation. The expression of novel miRNA *osa-novmiR3* was upregulated in the tolerant genotypes but downregulated in the sensitive rice genotypes (Appendix A). Its predicted target is granule-bound starch synthase 1b (Expectation value of 3.5 and UPE value of 17.32). It has an opposite expression in T1, T3, and T5 samples during vegetative growth (Figure 8C), indicating that granule-bound starch synthase 1b expression might be regulated negatively by novel miRNA *osa-novmiR3,* but the cause of downregulation of *GBSS1b* under low light stress in the tolerant rice genotypes is elusive. miRNA *osa-novmiR4* expression was found downregulated in the tolerant rice genotypes and upregulated in the sensitive rice genotypes except for the T1 sample (Appendix A). Its most predicted target is NAC domain-containing protein 48 (expectation value 2 and UPE value 19.39). Its expression was found upregulated in Swarnaprabha, whereas it was downregulated in the sensitive rice genotype IR8 (Figure 8D) indicating that its expression was also negatively regulated by novel miRNA *osa-novmiR4.* The NAC domain-containing protein shows enhanced leaf senescence at the grain-filling stage, which is expected to be involved in low-light molecular regulation [81]. Plants over-expressing NAC domain-containing protein 48 (NAC048) exhibit improved tolerance to drought and salt stress. It also improved tolerance to cold stress and resistance to rice blast fungus [82,83,84]. In this work, it has higher expression in tolerant genotypes under low light stress, and hence it might be involved in the regulation of the low-light-mediated tolerance mechanism. The expression of novel miRNA *osa-novmiR5* was also downregulated in the tolerant genotypes whereas it was upregulated in the sensitive rice genotypes (Appendix A). Its predicted target gene is FAD-dependent oxidoreductase and Cryptochrome2 apoprotein, whose expression was also analyzed in the tolerant and sensitive rice genotypes. Both FAD-dependent oxidoreductase and Cryptochrome2 apoprotein have a higher expression (Upregulated) in the tolerant genotype Swarnaprabha, whereas they were downregulated in IR8, indicating that it is negatively regulated by novel miRNA osa*-novmiR5* (Figure 8E,F). FAD-dependent oxidoreductase enzymes transfer electrons between the one-electron carrier ferredoxin and the two-electron carrier NADPH [85] and play a very important role in light-mediated energy molecule construction during photosynthetic pathways. Cryptochrome 2 apoprotein is a flavin-type blue light photoreceptor that regulates plant growth and development. It functions primarily under low light during the early development of seedlings in *Arabidopsis* [86]. Hence, Cryptochrome2 apoprotein has its role in the regulation of low-light-mediated responsive pathways. Its higher expression in tolerant rice genotypes enables it to cope with low light stress.

The above studies indicated that due to low light stress, changes occur in the genome-wide expression of different miRNAs that are involved in the regulatory pathways to regulate genes or transcript expression during stress to cope with it. The tolerant rice genotypes maintained photosynthesis and metabolites by enhancing the expression of positive regulators or suppressing genes that have the role as negative regulators mediated through different miRNA expressions. However, the sensitive rice genotypes were unable to maintain them. Therefore, we propose a model for the miRNA-mediated signaling pathway in low light stress tolerance during vegetative growth in rice (Figure 9).

## 4. Conclusions

Low light stress constitutes one of the most important environmental factors that hinder plant growth and development. MicroRNAs are potential regulators of gene expression in response to abiotic stresses. The up- or downregulation of miRNAs mediated by low light stress response primarily targets the genes for the photosynthetic machinery and metabolic pathways to adapt to the stresses. Our work provides the first small RNA expression analysis due to low light stress during vegetative growth in rice through NGS. Expression analysis and validation through *q*RT-PCR of known miRNAs *osa-miR166c-3p*, *osa-miR2102-3p,* and *osa-miR530-3p* and novel miRNAs *osa-novmiR1*, *osa-novmiR2*, *osa-novmiR3*, *osa-novmiR4*, and *osa-novmiR5* and their respective predicted targets show their involvement in low-light-mediated signaling mechanisms. Further study is required for a better understanding of the molecular pathways mediated through low light stress. Therefore, these identified miRNAs could be potential candidates for developing biotechnology-based low-light stress-tolerant rice varieties.

## 5. Experimental Procedures

### 5.1. Plant Material and Low Light Treatments

Two contrasting character rice genotypes, Swarnaprabha (low-light-tolerant) and IR8 (low-light-sensitive) were selected for the study. Selected rice genotypes were grown in the plant physiology net house of ICAR-National Rice Research Institute, Cuttack, Odisha, India. The low light (LL) treatment was simulated using an Agro shade net (75% of normal light) during vegetative growth (active tillering stage), 40 days after germination for 24 h (1 day), 72 h (3 days), and 120 h (5 days) and named as T1, T3, and T5, respectively, with control (C) receiving normal light (NL) in *Kharif* season 2018. Leaves were sampled from the main tillers of five different plants at the end of the treatment and five technical replicates were used for the experiments. The samples collected were first frozen in liquid nitrogen and then stored at −80 °C until use.

### 5.2. Photosynthetic Active Radiation (PAR) Measurement

The tolerant rice genotype Swarnaprabha and sensitive rice genotype IR8 were grown under low-light (LL) and normal-light (NL) conditions in the plant physiology net house of ICAR-National Rice Research Institute, Cuttack, Odisha, India in *Kharif* season 2018. The distribution of photosynthetic active radiation (PAR) above the canopy of plants under LL and NL conditions was recorded using a radiometer (LI-1500 LICOR, Nebraska, USA) thrice per day (9.00 am, 12.00 noon, and 4.00 pm in Indian Standard Time). Five replicates were recorded for each condition to measure accurate PAR value (Appendix A). For validation, three tolerant (Swarnaprabha, VLDhan-209, and Purnendu) and two sensitive (IR8 and GR4) genotypes were grown under low-light (LL) and normal-light (NL) conditions in *Kharif* season 2019. Similarly, the distribution of photosynthetic active radiation above the canopy of plants under LL and NL conditions was recorded in five replications (Appendix A). The Student’s *t*-test was used to know whether there was any significant difference ((*p* < 0.05, *p* < 0.01, and *p* < 0.01) in photosynthetic active radiation under NL and LL conditions.

### 5.3. Calculation of Chlorophyll Content and Photosynthetic Parameters

Chlorophyll content for NL- and LL-treated samples of Swarnaprabha and IR8 were estimated in the *Kharif* season of 2018. Similarly, Chlorophyll content for NL- and LL-treated samples of Swarnaprabha, VLDhan-209, Purnendu, IR8, and GR4 were estimated in the *Kharif* season 2019. Flag leaves from each of five replicates were taken, removed from their midrib, and sliced, and the sample was extracted with 10 mL 80% of acetone [63]. The extracts were stored in the dark for 24 h and then the absorbance was recorded at 663 nm and 645 nm using the colorimetric method to estimate the chlorophyll a and b contents in mg per gram fresh weight (mg g^−1^ FW) [87]. Five replicates for each sample were used for chlorophyll a and b content estimation (Figure 1A, C). Net assimilation, stomatal conductance, and transpiration measurements were recorded (Figure 1B, D) for both open system and treatment with a CO_2_ concentration of 380 μmol L^−1^ under available light conditions (Mu et al., 2010). The mean values for six flag leaves from each replicate were recorded to calculate the accuracy of the photosynthetic parameters in both the *Kharif* seasons of 2018 and 2019. The Student’s *t*-test was used to know whether there was any significant difference (*p* < 0.05, *p* < 0.01, and *p* < 0.01) in different physiological parameters between NL and LL conditions.

### 5.4. Total RNA Isolation, Library Preparation, and Small RNA Sequencing

Leaves were collected from each of three different plants of low-light-treated and control (untreated) samples. Three technical replicates were taken for total RNA isolation from four samples of Swarnaprabha (SC, ST1, ST3, ST5) and four samples of IR8 (IC, IT1, IT3, IT5) separately, using TRIZOL reagent (Invitrogen) following the manufacturer’s instruction, and the quality check of isolated RNA was performed with Qubit (pico green) to assess sample concentration. Then, an equal number of RNAs were pooled from three technical replicates for each sample, which was used for library preparation. The Illumina TruSeq Small RNA Library Preparation protocol was used to prepare the above-isolated RNA samples. Generally, mature miRNAs have 3’-hydroxyl and 5’-phosphate groups in response to the cellular pathway used to generate them. Due to this, the Illumina adapters are directly ligated to miRNAs in this kit. This protocol explains how to prepare libraries using total RNAs or purified small RNAs as input for subsequent cluster generation. The steps in this protocol were adapter ligation, reverse transcription, PCR amplification, and pooled gel purification to generate a library product. The adapters were ligated to the end of the RNA molecule and an RT reaction was used to generate single-stranded cDNA. The cDNA was then PCR-amplified using a primer containing 1 of 48 index sequences. The introduction of the index sequence at the PCR step separates the indexes from the RNA ligation reaction. This design allows for the indexes to be read using a second read and significantly reduces bias compared to designs that include the index within the first read. Illumina multiplexed sequencing uses 6-base indexes to distinguish different samples from each other in a single lane of a flow cell. Sequencing of all eight QC-passed libraries was performed on Illumina HiSeq 2500 System.

### 5.5. miRNA Sequence Analysis

The next-generation sequencing for small RNAs was performed using single-end (SE) 50 bp libraries on Illumina HiSeq 2500. The cutadapt version-1.10 was used for pre-processing of raw reads generated for the samples. Parameters used for cutadapt were minimum read length of 18 bp and maximum read length of 24 bp. Clean reads from each sample were provided to miARma-Seq tool Novel miRNA pipeline. miARma-Seq, which stands for miRNA-Seq and RNA-Seq multiprocess analysis, is a suite designed to study mRNAs, miRNAs, and circRNAs. It is able to perform differential expression analysis among others. Known and novel miRNAs were identified using miRDeep2 in miARma-Seq. The workflow for Illumina sequencing and bioinformatics analysis for the identification of miRNAs are shown in Appendix A.

### 5.6. Identification of Differentially Expressed miRNAs

The input data for differentially expressed miRNAs were read counts from miRDeep2 in miARma-Seq. Package edgR provided differential expression analyses of read counts arising from RNA-Seq. Hence, it was used for differential expression analysis of known and novel miRNAs. edgR analysis contains three steps: read count normalization, model-dependent *p*-value estimation, and FDR value estimation based on multiple hypothesis testing [88]. After stringent scrutiny, the read counts were trimmed to remove low-quality bases and length below 17 nt and above 24 nt, and with an FDR (false discovery rate) cutoff of ≤ 0.05. Trimmed mean of M values (TMM) was used for normalization, while DEGseq and HTseq [89,90] software were used for filtering the sequences. The *p*-value was adjusted for multiple testing with the Benjamini–Hochberg procedure, which controls the false discovery rate (FDR). Further differential screening of expressed miRNAs was performed as per standard padj < 0.05.

### 5.7. Expression Validation of Identified miRNAs

In *Kharif* season 2018, the same low-light-treated and control samples of Swarnaprabha and IR8 were used for validation of identified and selected miRNAs. In the next *Kharif* season of 2019, two more low-light-tolerant genotypes, Purnendu and VLDhan-209, and one more low-light-sensitive rice genotype, GR4 (selected based on our previous reports), were subjected to low light treatment along with Swarnaprabha and IR8 in a similar manner. The treated and control leaf samples were collected (three replicates) from three tolerant and two sensitive rice genotypes, frozen in liquid nitrogen and then stored at −80 °C until use. Total RNAs were isolated from all three replicates using PureLink Plant RNA Reagent (Invitrogen) following manual instructions. The quantitative RT-PCR was used for the validation of the expression of selected differentially expressed miRNAs and their predicted targets. Three known and five novel miRNAs were selected for validation in control and treated samples of tolerant and sensitive rice genotypes after NGS results analysis based on log_2_ (fold change) value, read count, and their involvement in photosynthetic and metabolic pathways.

### 5.8. Quantitative Real-Time PCR for Differentially Expressed miRNAs

*q*RT-PCR for the individual miRNAs was carried out using the kit miR-X qPCR (Clontech), and the steps followed were outlined in the instruction manual. Total RNAs isolated from the individual sample were polyadenylated and reverse-transcribed using poly (A) polymerase and SMART™ M-MLV reverse transcriptase, respectively. qPCRs were performed using SYBR^®^ Advantage^®^ qPCR Premix and mRQ 3′primer along with miRNA-specific and U6-specific 5′ primers and run on Bio-Rad CFX Manager Version 3.1 real-time PCR detection system. Delta-delta Ct method [91] was used for quantification of the miRNAs and the results were expressed as fold change of miRNAs in the samples from the low-light-treated and normal-light control plants. Primers for real-time PCR of miRNAs are shown in Appendix A. Results are represented in mean ± SD (standard deviation) of three replicates. The significance of differences (*p* ≤ 0.05) in the cellular abundance of miRNAs in the samples from the control and treated plants were tested by Paired *t*-test.

### 5.9. Target Prediction of Differentially Expressed miRNAs and Their Expression Validation through Quantitative PCR

Target predictions of selected differentially expressed novel and known miRNAs (Table 3) were obtained using the online available target prediction tool psRNATarget (http://plantgrn.noble.org/psRNATarget/ (accessed on 30 April 2021) [92]. The maximum expectation value (measures the complementarity between small RNA sequences and their target transcripts); hsp size (length for complementary scoring), and target accessibility were set at 5, 19, and 25, respectively. The range of central mismatch leading to translation inhibition was between 10 and 11 nt. Complete CDS of the individual target genes whose transcription was to be studied was blast-searched against the *Oryza sativa* L. sub sp. *indica* using the *Ensembl Plants* resource database (http://plants.ensembl.org/index.html (accessed on 30 April 2021) in order to verify the presence of the target sequence in the *indica* subspecies as well. Total RNAs were isolated from the leaves of control and low-light-treated samples of three tolerant (Purnendu, VLDhan 209, and Swarnaprabha) and two sensitive (GR4 and IR8) rice genotypes using PureLink Plant RNA Reagent (Invitrogen) and converted to cDNA using QuantiTect Reverse Transcription Kit (Qiagen). The kit provides gDNA Wipeout Buffer to remove genomic DNA contamination in total RNA isolated and has an optimized mix of oligo-dT and random primers to convert total RNA into cDNA. The cDNAs prepared from the individual samples were used for the study of the expression of the target genes. The expression of a gene was studied by qPCR, taking the cDNA as a template and SYBR green (Agilent). Primers specific to the gene of interest were designed using Primer Blast software at the NCBI site (Appendix A). The required amount of SYBR green, cDNA template, and the primers for a gene was mixed in a final volume of 20 µL, and PCR was run on Bio-Rad CFX Manager Version 3.1 real-time PCR detection system. Rice actin was used as an internal control. The relative levels of templates of the individual gene in the control and treated samples were quantified following Pfaffl [91], and the result was expressed as a fold change in the treated sample as compared to the control ones. Three technical replicates were used for each sample.

## Figures and Tables

**Figure 1 plants-11-02558-f001:**
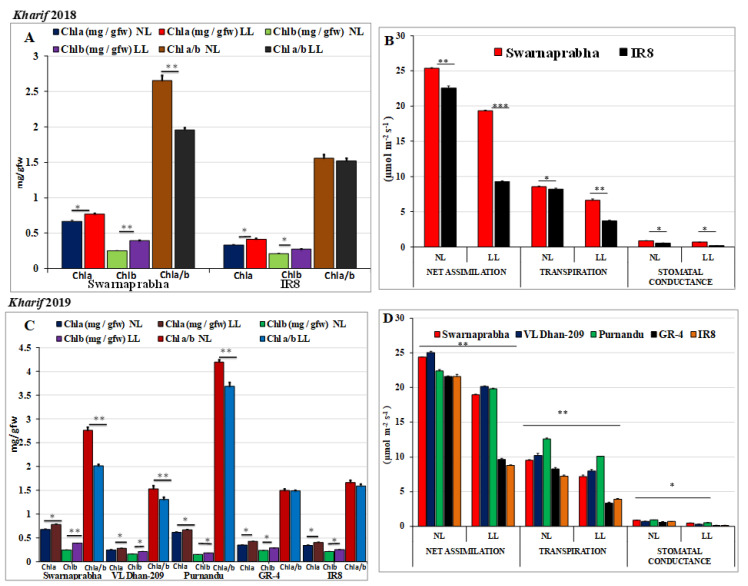
(**A**,**C**) Chla content, Chlb content, and Chl a/b ratio in different genotypes (*t*-test for mean data α = 0.05, 0.01, 0.001). Data were taken after 5 days of low-light-treated samples for both low-light-tolerant and low-light-sensitive rice genotypes in *Kharif,* 2018, and *Kharif,* 2019. Y-axis represents Chla, b content (mg/gfw) or Chla/b ratio. (**B**,**D**) Calculation of photosynthetic parameters in tolerant and sensitive rice genotypes in *Kharif* seasons of 2018 and 2019: Net assimilation, stomatal conductance, and transpiration measurements were recorded for both open system and treatment with a CO_2_ concentration of 380 μmol L^−1^ under available light conditions for LL—low light, NL—normal light. An average value was calculated from six flag leaves from each replicate. Note: In *Kharif,* 2018, only two rice genotypes, Swarnaprabha and IR8, were taken from the same samples which were used for NGS, whereas in *Kharif*, 2019, three tolerant genotypes—Swarnaprabha, VL Dhan209, Purnendu—and two sensitive rice genotypes—GR4, IR8—were taken for study. The student’s *t*-test (* *p* < 0.05, ** *p* < 0.01, and *** *p* < 0.001) was used to know whether there was any significant difference in Chla content Chlb content, Chla/b ratio, net assimilation, transpiration, and stomatal conductance between NL and LL conditions.

**Figure 2 plants-11-02558-f002:**
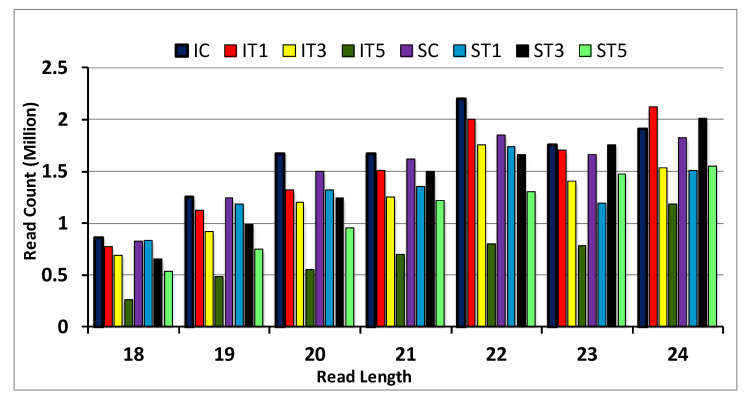
Read length (in nucleotide) distribution of miRNA (Illumina sequencing data) of Swarnaprabha (S) and IR8 (I) in low-light-treated T1, T3 T5, and control (C) samples. IC-IR8 control (untreated); IT1-IR8 for 1 day low-light-treated; IT3-IR8 for 3 days low-light-treated; IT5-IR8 for 5 days low-light-treated; SC—Swarnaprabha control (untreated); ST1—Swarnaprabha for 1 day low-light-treated; ST3—Swarnaprabha for 3 days low-light-treated; ST5—Swarnaprabha for 5 days low-light-treated.

**Figure 3 plants-11-02558-f003:**
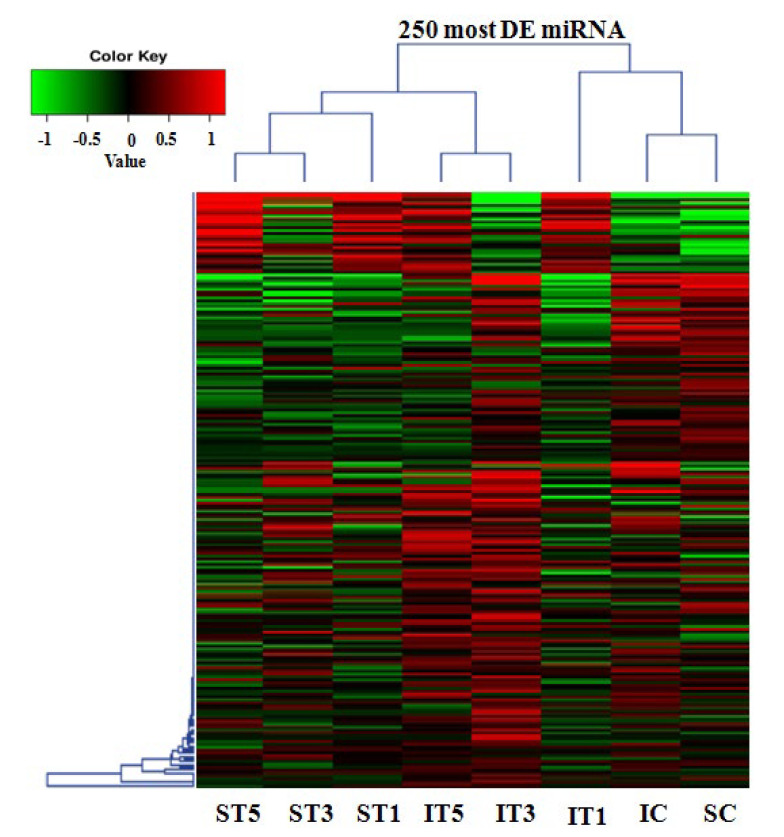
The heatmap of the 250 most differentially expressed (DE) miRNAs. Cluster analysis was performed using the log_2_(FPKM + 1) value from larger to smaller. The green color indicates downregulated miRNAs while the red color shows upregulated miRNAs. IC-IR8 control (untreated); IT1-IR8 for 1 day low-light-treated; IT3-IR8 for 3 days low-light-treated; IT5-IR8 for 5 days low-light-treated; SC-Swarnaprabha control (untreated); ST1-Swarnaprabha for 1 day low-light-treated; ST3-Swarnaprabha for 3 days low-light-treated; ST5-Swarnaprabha for 5 days low-light-treated.

**Figure 4 plants-11-02558-f004:**
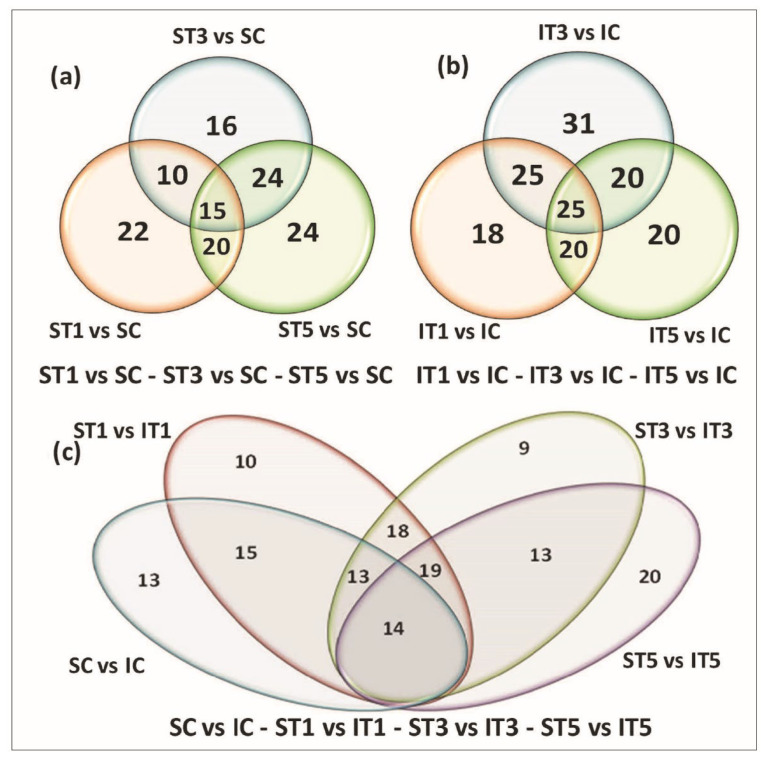
Venn diagram of differentially expressed miRNAs between different comparisons of tolerant (Swarnaprabha (S)) and susceptible (IR8 (I)) rice genotypes. The sum of numbers in each circle represents a total number of differentially expressed miRNAs within a group, while overlap represents the miRNAs expressed commonly in between comparisons. (**a**) Distribution of differentially expressed miRNA between ST1, ST3, and ST5 with respect to control (**b**) Distribution of differentially expressed miRNA between IT1, IT3, and IT5 with respect to control (**c**) Distribution of differentially expressed miRNA between SC, ST1, ST3, ST5 and IC, IT1, IT3, IT5. IC-IR8 control (untreated); IT1-IR8 for 1 day low-light-treated; IT3-IR8 for 3 days low-light-treated; IT5-IR8 for 5 days low-light-treated; SC—Swarnaprabha control (untreated); ST1—Swarnaprabha for 1 day low-light-treated; ST3—Swarnaprabha for 3 days low-light-treated; ST5—Swarnaprabha for 5 days low-light-treated.

**Figure 5 plants-11-02558-f005:**
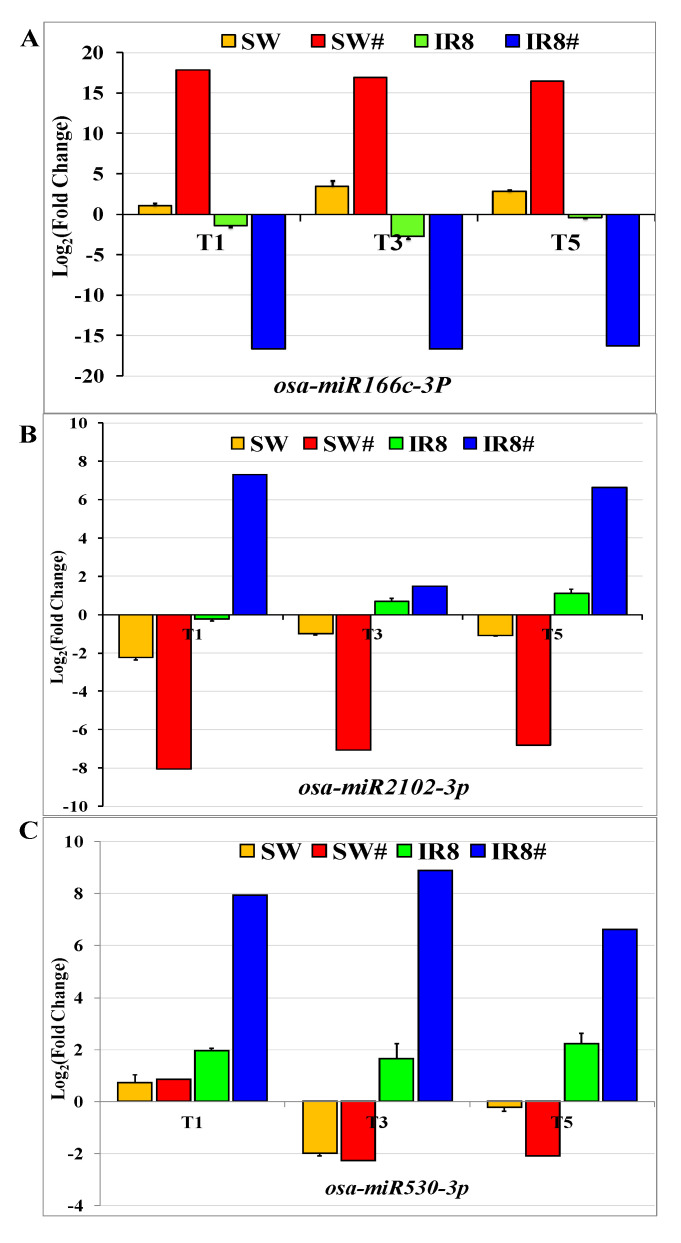
Validation of three differentially expressed known miRNAs, (**A**) *osa*-*miR166c-3p*, (**B**) *osa*-*miR2102-3p*, and (**C**) *osa*-*miR530-3p* through miRNA-specific qRT-PCR, identified from an miRNA sequencing results of Swarnaprabha and IR8 after low light treatment as compared to control. Each miRNA was amplified using mRQ 3′primer along with miRNA-specific and U6-specific 5′ primers. The primer sequences have been provided in Appendix A. (# = miRNA-seq). Error bars are ±SD of the average of three miRNA-specific *q*RT-PCR replicates.

**Figure 6 plants-11-02558-f006:**
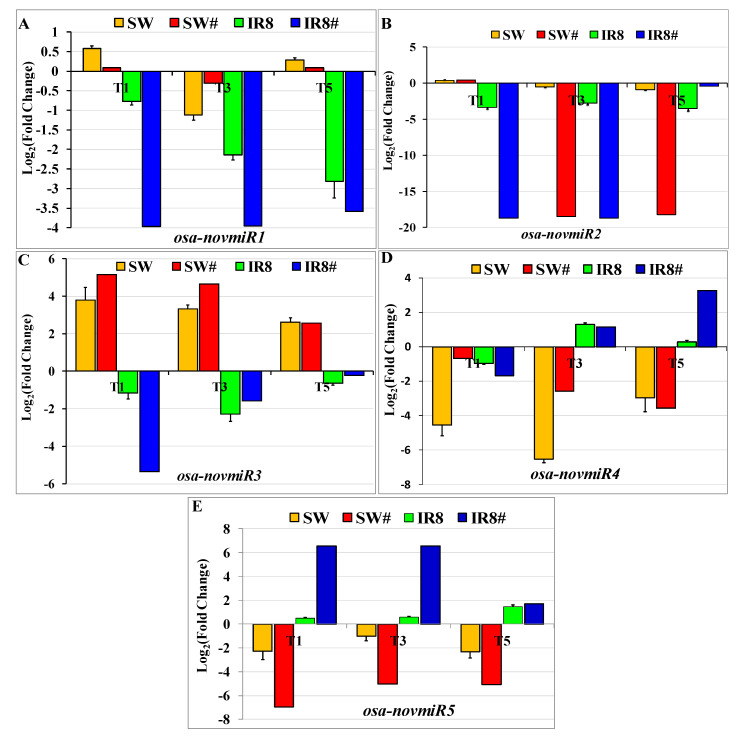
Validation of five differentially expressed novel miRNAs, (**A**) *osa*-*novmiR1*, (**B**) *osa*-*novmiR2*, (**C**) *osa*-*novmiR3*, (**D**) *osa*-*novmiR4*, and (**E**) *osa-novmiR5* through miRNA-specific *q*RT-PCR, identified from miRNA sequencing results of Swarnaprabha and IR8 after low light treatment with respect to control. Each miRNA was amplified using an mRQ 3′primer along with miRNA-specific and U6-specific 5′ primers. The primer sequences are provided in Appendix A. (# = miRNA-seq). Error bars are ±SD of the average of three miRNA-specific *q*RT-PCR replicates.

**Figure 7 plants-11-02558-f007:**
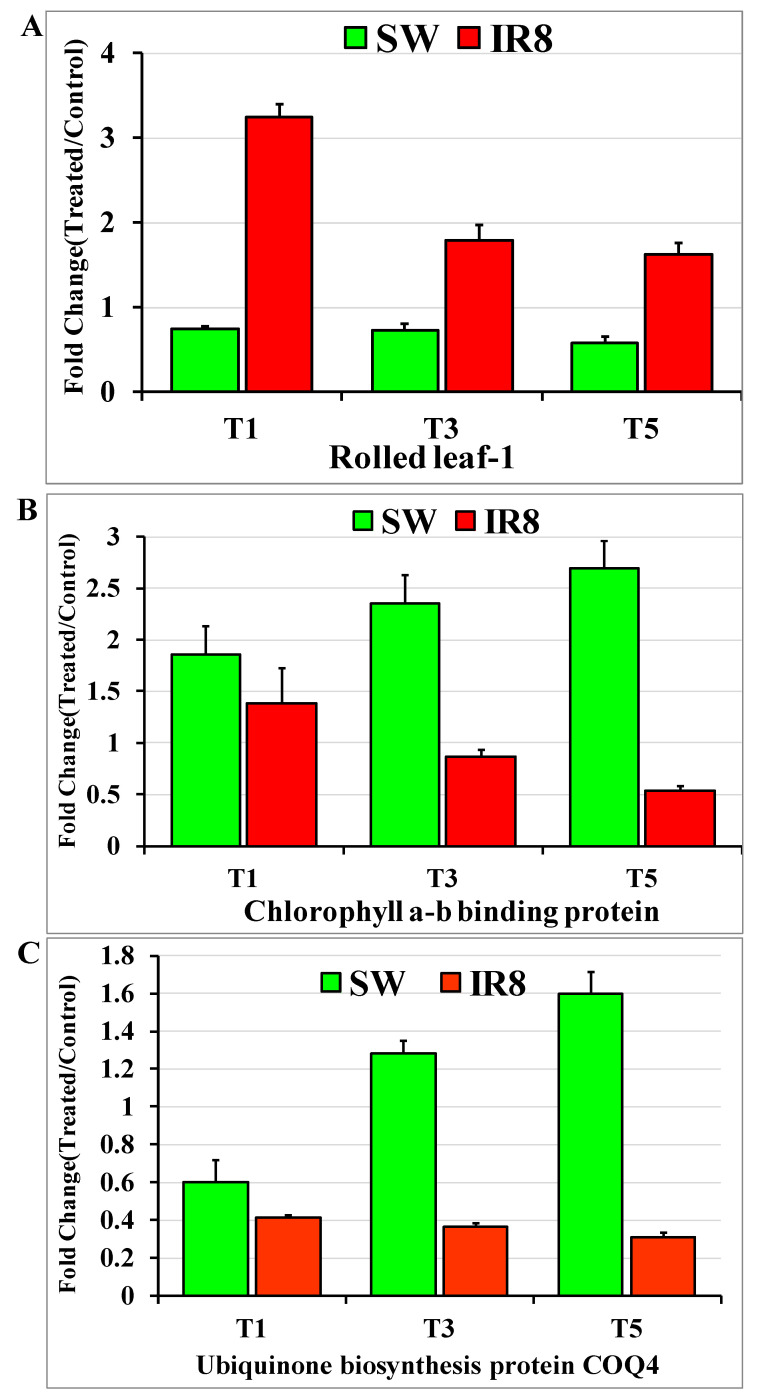
Expression analysis of target genes of identified three known miRNAs in T1, T3, and T5 samples of Swarnaprabha (SW) and IR8 with respect to control. (**A**) Rolled leaf-1 expression, a target of miRNA *osa-miRNA166c-3P*, (**B**) chlorophyll a-b-binding protein, a target of miRNA *osa-miR2102-3p*, (**C**) ubiquinone biosynthesis protein COQ4, a target of miRNA *osa-miR530-3p.* Each gene was amplified using gene-specific primers designed using Primer Blast Tool. Actin was taken as an internal positive control. The primer sequences are provided in Appendix A. Error bars are ±SD of the average of three *q*RT-PCR replicates.

**Figure 8 plants-11-02558-f008:**
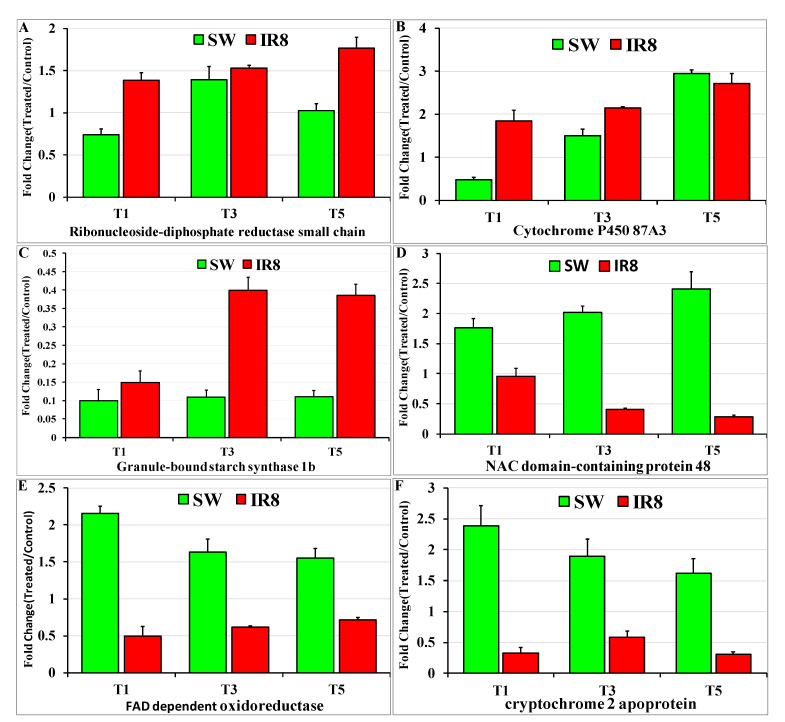
Expression analysis of target genes of identified five novel miRNAs in T1, T3, and T5 samples of Swarnaprabha (SW) and IR8 with respect to control. (**A**) Ribonucleoside-diphosphate reductase expression, a target of miRNA *osa-novmiR1*; (**B**) cytochrome P450 87A3, a target of miRNA *osa-novmiR2*; (**C**) *GBSS1b,* a target of miRNA *osa-novmiR3;* (**D**) NAC domain-containing protein, a target of miRNA *osa-novmiR4;* (**E**) FAD-dependent oxidoreductase, a target of miRNA *osa-novmiR5;* and (**F**) cryptochrome2 apoprotein, a target of miRNA *osa-novmiR5.* Each gene was amplified using gene-specific primers designed using Primer Blast. Actin was taken as an internal positive control. The primer sequences are provided in Appendix A. Error bars are ±SD of the average of three qRT-PCR replicates.

**Figure 9 plants-11-02558-f009:**
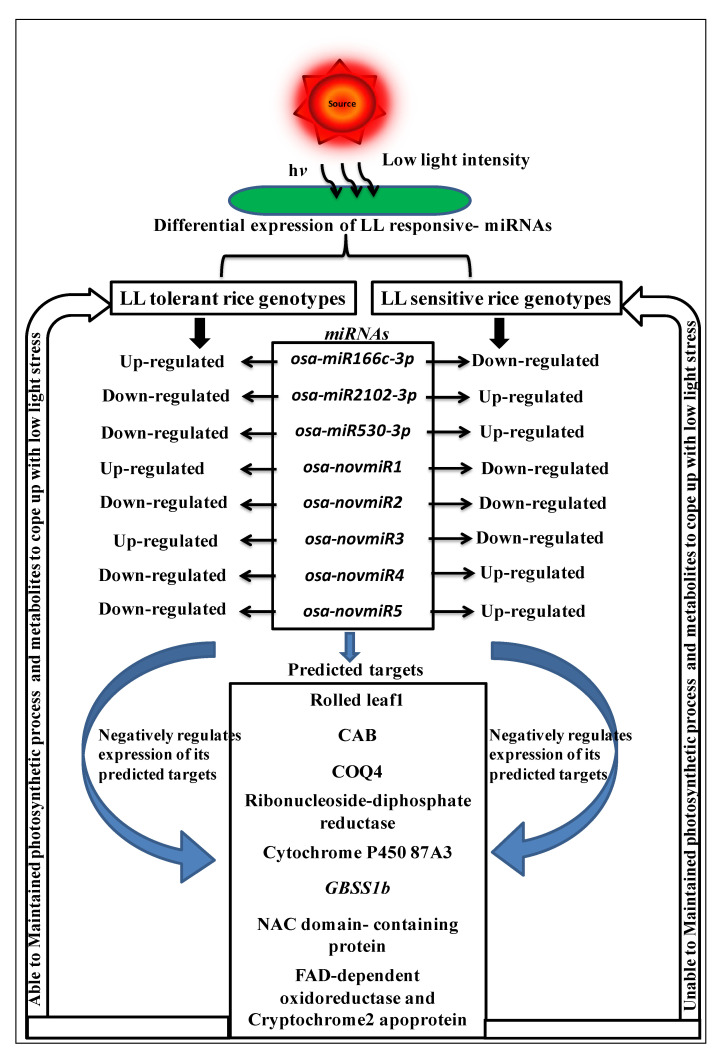
Proposed model depicting miRNA-mediated tolerance or susceptibility due to low light intensity stress in low-light-tolerant and -sensitive rice genotypes. Among identified differentially expressed miRNAs in tolerant and sensitive rice genotypes, three known and five novels were selected. Upregulated miRNAs show downregulation of their respective targets and vice versa. Results indicate that miRNAs are expressed in such a way that their respective targets’ expression involved maintaining photosynthetic and metabolic pathways to cope with low light stress in tolerant rice genotypes, but sensitive rice genotypes were unable to maintain it.

**Table 1 plants-11-02558-t001:** Basic statistics of total raw and clean reads of small RNA sequencing (miRNA) data for control and low-light-treated samples of rice genotype Swarnaprabha (low-light-tolerant) and IR8 (low-light-sensitive).

Sample Name	Raw Reads	Raw Bases (GB)	Clean Reads	Error Rate (%)	Q20%	Q30%	GC Content %
SC	41,388,259	2.069	10,431,749	0.01	98.56	97.42	53.14
ST1	48,021,253	2.401	9,245,077	0.01	98.37	97.08	52.83
ST3	42,368,691	2.118	9,723,848	0.01	98.51	97.38	53.29
ST5	43,507,150	2.175	7,849,060	0.01	98.52	97.38	53.70
IC	47,126,461	2.356	11,174,458	0.01	98.49	97.30	53.18
IT1	50,061,687	2.503	10,417,005	0.01	98.58	97.49	53.05
IT3	33,181,938	1.659	8,644,198	0.01	97.61	94.78	52.39
IT5	34,541,807	1.727	4,929,964	0.01	97.40	94.12	52.87
Total	340,197,246	17.008	72,415,359	-	-	-	-
Average	42,524,655.8	2.13	9,051,919.88	0.01	98.255	96.62	53.18

SC—Swarnaprabha control, ST1—Swarnaprabha for 1 day low-light-treated; ST3—Swarnaprabha for 3 days low-light-treated; ST5—Swarnaprabha for 5 days low-light-treated; IC—IR8 control; IT1—IR8 for 1 day low-light-treated; IT3—IR8 for 3 days low-light-treated; IT5—IR8 for 5 days low-light-treated.

**Table 2 plants-11-02558-t002:** Differentially expressed miRNAs statistics identified through small RNA NGS data analysis.

Comparisons between Samples	No of Upregulated miRNAs	No of Downregulated miRNAs	No of Differentially Expressed miRNA
IT1 vs. IC	53	35	88
IT3 vs. IC	62	39	101
IT5 vs. IC	53	32	85
ST1 vs. SC	33	34	67
ST3 vs. SC	28	37	65
ST5 vs. SC	26	57	83
SC vs. IC	48	33	81
ST1 vs. IT1	44	50	94
ST3 vs. IT3	33	52	85
ST5 vs. IT5	30	70	100

SC—Swarnaprabha control, ST1—Swarnaprabha for 1 day low-light-treated; ST3—Swarnaprabha for 3 days low-light-treated; ST5—Swarnaprabha for 5 days low-light-treated; IC—IR8 control; IT1—IR8 for 1 day low-light-treated; IT3—IR8 for 3 days low-light-treated; IT5—IR8 for 5 days low-light-treated.

**Table 3 plants-11-02558-t003:** List of differentially expressed miRNAs identified through small RNA sequencing selected for validation.

Sl. No.	miRNA ID	Mature miRNA Sequences	Type of miRNA	Type of Inhibition	Target
1	*osa-miR166c-3p*	UCGGACCAGGCUUCAUUCCCC	Known	Cleavage	Rolled leaf1
2	*osa-miR2102-3p*	CGGGGCCGGUUCCGGUGUAGG	Known	Translation	Chlorophyll a-b-binding protein
3	*osa-miR530-3p*	AGGUGCAGAGGCAGAUGCAAC	Known	Translation	Ubiquinone biosynthesis protein COQ4/ oxidoreductase
4	*osa-novmiR1*	AGCUCGUCGGGCUUGCUGCGG	Novel in rice	Cleavage	Ribonucleoside-diphosphate reductase
5	*osa-novmiR2*	AAGUCCUCGUGUUGCAUCCCU	Novel in rice	Cleavage	cytochrome P450 87A3
6	*osa-novmiR3*	UGCCGGUCAUAUGUAUCGAA	Novel in rice	Translation	Granule-bound starch synthase 1b
7	*osa-novmiR4*	ACCGCUUCAUGAACUUUCAGG	Novel in rice	Cleavage	NAC domain-containing protein
8	*osa-novmiR5*	CAAAUCCUGUCAUCCCUACC	Novel in rice	Cleavage	FAD-dependent oxidoreductase and Cryptochrome 2 apoprotein

## Data Availability

Not applicable.

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
