# Peer review of "Identification of microRNAs That Provide a Low Light Stress Tolerance-Mediated Signaling Pathway during Vegetative Growth in Rice"

_plants, 2022, doi:10.3390/plants11192558_

Round 1

Reviewer 1 Report

Authors present really interesting work, that is novel and useful. Everything is great besides one fact - samples were take from only one plant per treatment. Nowadays at least three independent samples or in worst case three pooled samples in library are acceptable.

Authors tried to solve this by repeating some analysis by qPCR, but even here it seems sample were collected only from one plant.

Also, authors claim to use low-light tollerant and low-light sensitive lineages, but there is no characterisation based on which are these lineages selected (e.g. some gene/seed bank data, references to light intensity experiments etc.).

Also quality of Figure 3 is terrible, and there is no explanation of the clustering method that was used (and the cluster categories on the miRNA axes).

Author Response

Query 1.  Authors present really interesting work that is novel and useful. Everything is great besides one fact - samples were taken from only one plant per treatment. Nowadays at least three independent samples or in worst case three pooled samples in library are acceptable.

Compliances

Samples have been collected from the thee independent plants at the end of treatment, and it were mentioned in the manuscript. Further, three technical replicates for biological samples were taken for mRNA isolation. Then, an equal amount of RNAs was pooled from three technical replicates for each sample, which was used for library preparation.

Query 2.  Authors tried to solve this by repeating some analysis by qPCR, but even here it seems sample were collected only from one plant.

Compliances

Samples were collected from three different plants with three replicates for mRNA isolation.

Query 3.  Also, authors claim to use low-light tolerant and low-light sensitive lineages, but there is no characterization based on which are these lineages selected (e.g. some gene/seed bank data, references to light intensity experiments etc.).

Compliances

Low-light tolerant and low-light sensitive lineages were selected based on the  previous works and our experiments (Nayak and Murty, 1980, Sekhar et al., 2019, Kumar et al., 2019, Ganguly et al., 2020; Kumar et al., 2020. We have screened 300 genotypes during 2016 and 2017 under low light and normal light conditions at three different places of India. We have identified few low light tolerant and several susceptible rice genotypes based on the Agro-morphological and yield related traits. Few tolerant and susceptible genotypes were further studied for different physiological and biochemical traits and some repots have been published.

References

Nayak, S.K.; Murty, K.S. (1980). Effect of varying light intensities on yield and growth parameters in rice. Indian J. Plant Physiol., 23(3), 309-316.

Kumar, A., Panda, D., Biswal, M.,  Dey, P., Behera, L., Baig, M.J.,  Nayak, L., Ngangkham, U., Sharma, S.G. (2019). Low light stress influences resistant starch content and glycemic index of rice (O. sativa L). Starch‐Stärke,  71( 5-6), 1800216. 

Sekhar, S., Panda, D., Kumar, J., Mohanty, N., Biswal, M., Baig, M.J., Kumar, A., Ngangkham, U., Samantaray, S., Pradhan, S.K., Shaw, B.P., Behera, L. (2019). Comparative transcriptome profiling of low light tolerant and sensitive rice varieties induced by low light stress at active tillering stage. Sci. Rep., 9, 5753.

Ganguly, S.; Saha, S.; Vangaru, S.; Purkayastha, S.; Das, D.; Saha, A.K.; Roy, A.; Das, S.; Bhattacharyya, P.K.; Mukherjee, S.; et al.(2020). Identification and analysis of low light tolerant rice genotypes in field condition and their SSP-based diversity in various abiotic stress tolerant line. J. Genet., 99, 24–33.

Kumar, A., Panda, D., Mohanty, S., Biswal, M.,  Dey, P., Dash, M.,  Prasad Sah, R.P., Sudhir Kumar, S., Baig, M.J.,  Behera, L. (2020). "Role of sedoheptulose-1, 7 bisphosphatase in low light tolerance of rice (Oryza sativa L.)." Physiol. Mol. Biol. Plants, 26(12),  2465-2485.

Query 4. Also quality of Figure 3 is terrible, and there is no explanation of the clustering method that was used (and the cluster categories on the miRNA axes).

Compliances

FPKM cluster analysis was done using the log2 (FPKM+1) value.

Reviewer 2 Report

In the present work, 513 new miRNAs in rice were identified from the in total 9 million clean reads. The content presented in the paper is sufficient, but needs more discussion about the role of the new miRNAs found in low light intensity. The results of the study did present the particularity of the rice species during light stress.

I would suggest to update the references in the introduction section because they are quite old.

I think Table 1 has a mistake. The total % in Q20 (785 %), Q30 (772%) and GC (424%) is not correct. Check all the data to confirm.

Quality of figures 2, 3 can be improved.

The model of figure 9 can be improved. It is difficult to follow.

The references in the discussion section are quite old. There are new important published information that might be discussed.

It is not clear how many biological and technical replicates were used

Author Response

Query 1.  In the present work, 513 new miRNAs in rice were identified from the in total 9 million clean reads. The content presented in the paper is sufficient, but needs more discussion about the role of the new miRNAs found in low light intensity. The results of the study did present the particularity of the rice species during light stress.

Compliances

Added some more information about new miRNAs in the discussion part.

Query 2.  I would suggest to update the references in the introduction section because they are quite old.

Compliances

New references have been added in the introduction.

Query 3.  I think Table 1 has a mistake. The total % in Q20 (785 %), Q30 (772%) and GC (424%) is not correct. Check all the data to confirm.

Compliances

Mistakenly it was also added with other data. I have removed it from the Table 1

Query 4. Quality of figures 2, 3 can be improved

Compliances

We have improved it as per possible and added in the manuscript

Query 5. The model of figure 9 can be improved. It is difficult to follow.

Compliances

Improvement has been done. A note has been written below the figure.

Query 6. The references in the discussion section are quite old. There are new important published information that might be discussed

Compliances

New references have been added in the discussion section. Few important information have been discussed.

Query 7. It is not clear how many biological and technical replicates were used

Compliances

Samples have been collected from the three independent plants at the end of treatment. Further, three technical replicates for biological samples were taken for mRNA isolation. Then, an equal amount of RNAs was pooled from three technical replicates for each sample, which was used for library preparation.

.

Reviewer 3 Report

The manuscript used microRNA sequencing to profile and identify differentially expressed microRNAs in response to low light stress in rice. The target genes of several differentially expressed microRNAs were also examined in this study. The topic of this study is very interesting and significant, the experiments were well designed and results were properly presented. This study has provided some valuable informations for low light study. Here are some concerns about this manuscript.

1. It will be better to make the title of this manuscript more concise.

2. In figure 1a and 1c,no description for y axis, the authors can used double y axis for these figures.

3. Is there any significant difference in these photosynthesis parameters between NL and LL conditions in Figure 1b and 1d? If yes, please do the significant difference analysis and indicate in the figure.

4. In figure 5 and 6, the expression results for these 8 selected microRNAs should be presented in the main result or as a supplement figure to show their consistence with qRT-PCR results.

5. Statistic analysis for significant difference has been used in figure 1, however, the relative method was not described in the methods part.

Author Response

Query 1.  It will be better to make the title of this manuscript more concise.

Compliances

Title has been changed as per comment.

Query 2.  In figure 1a and 1c,no description for y axis, the authors can used double y axis for these figures.

Compliances

Y-axis has been described under note in  the figure 1a and 1c.

Query 3.  Is there any significant difference in these photosynthesis parameters between NL and LL conditions in Figure 1b and 1d? If yes, please do the significant difference analysis and indicate in the figure.

Compliances

The student's T-test (* P < 0.05, ** P < 0.01, and ***  P < 0.001) was used to know whether there was any significant difference in the physiological parameters  between NL and LL conditions.

Query 4. In figure 5 and 6, the expression results for these 8 selected microRNAs should be presented in the main result or as a supplement figure to show their consistence with qRT-PCR results.

Compliances

The expression results for 8 selected microRNA already explained in the results part of the manuscript

Query 5. Statistic analysis for significant difference has been used in figure 1, however, the relative method was not described in the methods part.

Compliances

The student's T-test (* P < 0.05, ** P < 0.01, and  ***  P < 0.001) was used to know whether there was any significant difference in the physiological parameters  between NL and LL conditions.
